# Support potential of elite civil universities for China's space industry: Higher educational mobilization capacity for China's space ambition

Xiaoxiao Li [1,2]*, Wei Niu [2]

**1** Department of Education and Research on National Defense, Fuzhou University, Minhou County, Fuzhou, P. R. China, **2** Beijing Kaiyuan Performance Management and Evaluation Service Center, Donggaodi, Fengtai, Beijing, P. R. China

* shawnleemm@gmail.com, shawnlee@fzu.edu.cn

## Abstract

This study established a model to assess the capability of civil universities to support the development of China's space industry. Using linear scaling, data from 41 elite universities were assessed for three dimensions: education, research, and culture. Differences among the indicators and their correlations were examined. A k-means algorithm was employed to classify universities into four clusters. According to the results, high support potential for the space industry is concentrated in certain universities. With the outcomes, the study also provides recommendations to guide space industry authorities and university leaders in policy-making. Further, the model can be used as a feasible, simple, and practical tool for similar studies.

## 1. Introduction

In Chinese, the word *spaceflight* (*Hangtian*) refers not only to satellite launching, manned space flight, and scientific exploration but, more importantly, also to missile and missile-related weaponry systems, research, and development. Since the establishment of the space industry in October 1956, its most important mission has been ballistic missile system development [1]. The space industry undertakes research, development, and mass production of numerous classes of ballistic and cruise missiles, which is an advanced technological grouping containing more than 40 types that can carry either conventional or nuclear warheads [2]. This missile arsenal may be one of the most important martial strengths of the People's Liberation Army. Owing to the Intermediate-Range Nuclear Forces Treaty, China has taken the lead over the US in developing intermediate-range ballistic missiles in recent years [3]. Additionally, the space industry is developing new concept weapon systems, such as the glide hypersonic missile, stealth drones, and near-space vehicles, which could change the military balance in the Indo-Pacific region [3, 4]. In fact, the space industry has been a strategic sector for China; its development has been critical to fulfilling Beijing's ambitions for great power and now represents the cutting edge of a larger technological development [5]. Scholars,

**Data Availability Statement:** The minimal anonymized raw data of our study are available on the Open Science Framework website (DOI: 10.17605/OSF.IO/HVZ2Q).

**Funding:** This study was supported by grant number 2014M561162 from the 56th Chinese Postdoctoral Science Foundation and grant number CXRC201915 from the Fuzhou University Research Start-up Fund. The funders had no role in study design, data collection and analysis, decision to publish, or preparation of the manuscript.

**Competing interests:** The authors have declared that no competing interests exist.

politicians, and military officers around the globe should give increased attention to China's space power development to understand the capability-generating mechanism of the space industry. The space industry mobilization potential in China is especially meaningful for peacekeeping within the region and around the world.

Because of the low transparency of the space industry, it is very difficult to obtain relevant details in China. However, monitoring HR activities can provide some understanding of the industry's relationship with universities. The nexus between higher education and the space sector may contribute significantly to the sustainable development of the space industry; yet, this relationship has not been evaluated qualitatively. It is crucial to establish a comparable and reliable index to measure the supporting potential of elite universities for the defense industry—especially, the space industry in China. A robust understanding of how universities can assist the space industry by providing HR and research and development (R&D) collaboration would be valuable to corporate executive managers in this industry.

In 2018, China's public spending on education exceeded three trillion yuan [6]. It is interesting to note that when China's defense spending increases, education expenditures also increase [7]. This dynamic implies that a state plan for the university system can include its potential to support areas of the defense sector. Among China's state university investment plans, the 211 and 985 Projects are the most important. The 985 Project, first announced on 4 May 1998, is a long-term state plan to promote China's higher education system by founding world-class universities in the 21st century [8]. The 211 Project, initiated in November 1995, aims to raise educational standards in approximately 100 colleges and universities in the 21st century [9]. The 985 Project endeavors to select the cream of the crop from the 211 Project. Thus, the 985 Project-listed universities constitute the core elite universities in China and are considered as the country's "Ivy League" schools. However, the supporting potential of all universities—for a particular area of the strategic industry—has not been assessed.

China's elite universities have a complex orientation, and thus their characteristics differ. The universities founded before 1949 were fundamentally restructured according to their departments, and organized according to professional discipline [10, 11]; meanwhile, a series of new universities has been established, of which science and engineering universities (SEUs) account for a significant proportion. From 1950 to 1960, 47 universities were established, and later included in the 211 and 985 Projects, among which 46.81% were SEUs. Some SEUs offer a particular industry direction. For example, since 1965, China has established eight ministries of machine-building (MMBs). Some of the MMBs have their own affiliated SEU [12]. In most cases, each SEU has its own unique direction, through which it supports a given defense industry sector. When China entered the era of "reform and opening up," the MMBs were reorganized into general industry corporations. In 1999, seven significant affiliated SEUs shifted from general industry corporations to State Commissions of Science and Technology for the National Defense Industry [12]. Those seven SEUs are currently affiliated with the Ministry of Industry and Information Technology under the central government [13] and included in the 211 or 985 Projects.

After the change in administrative subordination in 1999, the seven SEUs and other superior universities were developed over the following two decades according to the state plans. On one hand, the guiding principles of the seven SEUs had changed over time, allowing each to potentially serve one or more defense industry sectors. On the other hand, other elite universities could intentionally or otherwise develop strategies and capabilities to support space industry sectors. Given this background, the answers to the following questions would be useful for the defense industry authorities to facilitate potential industrial mobilization in the space industry: Which university is most advantageous, or potentially so, for the space industry? What type of support could a university offer to the sector? Based on the answers to these

questions, some recommendations could be provided to the industry administration and university leadership that could help them improve managed strategizing, HR education and recruitment, and research collaboration. Moreover, it could help observers understand the nature of the production of intellectual capital in the industry, and the potential for relationship between the industry and the university system.

The objectives of this study are as follows: 1) to discover the HR needs and structure of China's space industry at the disciplinary level, 2) to develop an index system to identify the universities that can render advanced comprehensive support to China's space industry, 3) to explore the differences between the former MMB-affiliated universities and other high-quality universities listed in the 985 Project, and 4) to examine the relationships among the space industry-supporting indicators and explain their underlying mechanisms.

## 2. Theoretical framework

Some scholars argue that intellectual capital is the source of core competencies for enterprises [14]. Other scholars believe knowledge growth enhances the management and delivery practices of organizations, helping them better prepare for the future [15]. Studies have proven that a definite relationship exists between intellectual capital and organizational innovation [16, 17]. The space industry is a highly technical R&D industry, characterized by knowledge-intensive activities. Therefore, intellectual capital is integral to the industry's development, and the university system could contribute to the acquisition of this crucial resource.

Intellectual capital theory can be traced back to the end of the 20th century, when Edvinsson divided intellectual capital into human and structural capital [18]. By comparing theories posited by previous scholars, Viedma separated intellectual capital into three categories: human, structural, and relational [19]. According to most scholars, human capital is the foundation of intellectual capital; structural capital is the worth and value created within an organization that remains when employees leave the organization, and relational capital is the knowledge embedded in relationships with customers, strategic partners, suppliers, distributors, investors, public bodies, and other stakeholders [17, 20–22]. From relational capital, some scholars derived the concept of external intellectual capital, defined as the sum of cooperative organizations' or individuals' knowledge, skills, innovation ability, cultures, processes, and other intangible assets that can be applied directly or indirectly by the organization [17].

In China, the fundamental university functions are delineated as HR education, scientific research, and social services [23]. The university system could supply HR to the space industry, which enriches the core human capital and provides space science and technology, knowledge, and management solutions that support the structural capital. A university could be a valuable cooperative partner that broadens the industry's external intellectual capital. Accordingly, universities could be considered the most effective producers and suppliers of intellectual capital for the space industry. To assess the supportive potential of universities for the space industry, this study focused on the intellectual capital production capabilities of high-quality universities in China.

Strategic management concepts can enable university leaders to identify problems, formulate strategies, and determine the compulsory capabilities of leaders to respond to challenges and achieve strategic goals [24]. Accordingly, the functionality of the university support mechanism may rely on strategic management. Therefore, some elements of strategic management models [25], such as innovative culture, must be considered. Organizational culture includes, in the least, an organization's expectations, experiences, philosophy, and the values that guide member behavior. These elements are expressed in members' self-image, inner workings, interactions with the outside world, and future expectations [26]. The organizational culture

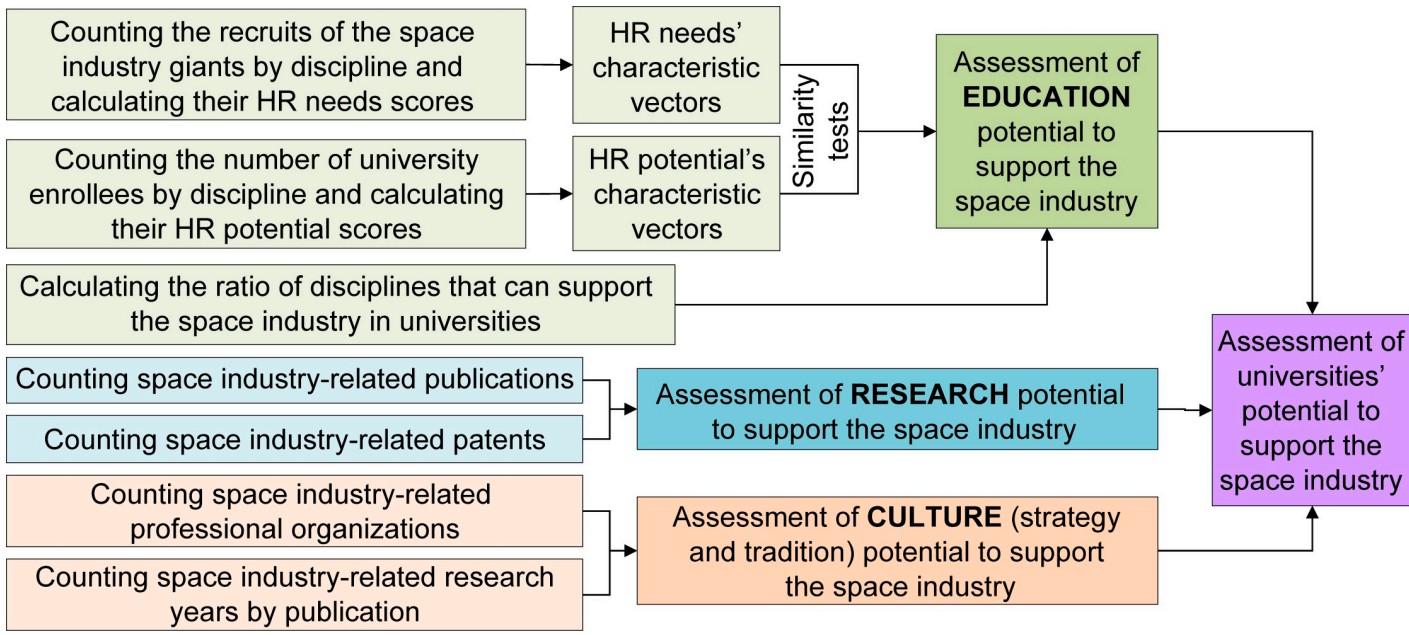

**Fig 1. Theoretical framework of index development.**

in modern higher educational institutions has a significant influence on the nurturing of "human potential," "human capital," and "intangible assets" [27]. Although numerous aspects could reflect the domain, this study selected leadership strategy and faculty research traditions as representatives.

Therefore, the supporting potential of universities for the space industry was evaluated, considering three domains: education, research, and culture. Further, the three domains were divided into a series of subordinate indicators (Fig 1).

## 3. Materials and methods

In the 21st century, China's space industry is concentrated in two giant corporations: the China Aerospace Science and Technology Corporation (CASC) and China Aerospace Science and Industry Corporation (CASIC) [28]. The former has 8 large research and development (R&D) and production complexes, 11 specialized companies, 13 listed companies, and several directly affiliated units; the latter has six research institutes and 17 wholly owned holding companies and subordinate units; each has more than 150 thousand employees [29, 30]. Today, both CASC and CASIC have become Fortune Global 500 firms [29, 30]. These two defense industry giants dominate China's space industry in the fields of launch vehicles, space crafts, satellites, ballistic missiles, hypersonic near-space vehicles, and so on. Their HR policy is in high accordance with the needs of engineering projects and further emphasizes HR's excellence, reserve, training, and team formation. In recent years, the corporations have posted approximately three to four thousand graduate job vacancies aimed at China's best universities [29–31]. Thus, CASC's and CASIC's HR needs were considered by this study, as representative of the requirements of China's space industry.

As a socialist country, most universities in China are owned and run by the state and have a direct affiliation with the government. Thus, in this study, universities that are affiliated with the civil branch of China's central or local government and that are under direction by the governments' civil educational sector are defined as a civil university. The civil universities listed

in the 985 Project and former MMB universities were included in this study as research subjects. Hence, a total of 41 universities were included as samples for assessment. These universities are state-run civil, comprehensive universities and are directly subordinate to central government ministries; none are subordinate to lower-level governments or private establishments.

The campus recruitment data for CASC and CASIC were collected and published online, including recruitment, advertisements, and publicity material. The raw data included information on the job category, discipline, academic degrees, and recruitment numbers [31]. Overall, 25,627 job requirements were retrospectively collected from 2014 to 2019 (Table 1).

The enrollment numbers for each discipline (2017 to 2019) and the statuses of the professional organizations (college and research organizations at all levels) from the universities' websites in September 2019 were collected. The publication and patent data were derived primarily from China's National Knowledge Infrastructure (CNKI) database, which is considered comprehensive [32]. The publication and patent data were retrieved before September 2019. The CNKI has developed its own English publication-retrieving function; thus, English publications were also retrieved. Additionally, the ISI Web of Knowledge was employed as a supplement.

There were 1,388,293 enrollment cases collected among the 41 universities. The annual average case numbers of bachelor's, master's, and doctorate enrollments in a university were 4985.23 ± 1860.20, 5109.05 ± 1657.32, and 1192.66 ± 626.58, respectively. Data from 5,544 articles, 3,468 conference papers, 2,262 doctoral dissertations, 8,226 master's theses, and 4,358 patents were retrieved from the database.

## 4. A pilot study

### 4.1 Creation of a disciplinary classification system

First, to enhance the comparability of universities' HR potential and the space industry's HR needs at different education levels (bachelor's, master's, and doctoral), the disciplines in China's higher education system were standardized based on a comprehensive three-tier coded classification system, which was developed with the assistance of an expert panel based on the Ministry of Education's classification at each educational level [33, 34]. According to the coded classification system, we classified all disciplines' titles in raw data into appropriate coded titles for analysis. Table 2 provides a brief description of the disciplinary classification system in its first-tier discipline's title and the number of second- and third-tier disciplines under each first-tier discipline.

**Table 1. Description of collected data of job requirements.**

| Employer corporation | Annual job posting | Number of jobs required | Discipline requirement proportion (%) | | | | Degree requirements (%) | | |
|---|---|---|---|---|---|---|---|---|---|
| | | | Engineering | Natural science | Social science | Others | Bachelor | Master | Doctor |
| CASC | 2015 | 3363 | 88.49 | 4.10 | 5.98 | 1.43 | 22.48 | 61.71 | 15.81 |
| | 2016 | 3091 | 91.94 | 1.20 | 5.60 | 1.26 | 17.76 | 68.78 | 13.46 |
| | 2017 | 3075 | 91.80 | 1.17 | 6.08 | 0.94 | 14.76 | 73.07 | 12.16 |
| | 2019 | 3457 | 91.90 | 1.33 | 5.79 | 0.98 | 20.16 | 71.71 | 8.13 |
| CASIC | 2014 | 4961 | 89.91 | 3.86 | 5.92 | 0.30 | 52.29 | 43.72 | 3.99 |
| | 2015 | 3162 | 90.94 | 3.44 | 5.51 | 0.11 | 40.26 | 50.00 | 9.74 |
| | 2018 | 4518 | 89.05 | 4.08 | 6.82 | 0.06 | 17.02 | 68.81 | 14.17 |

CASC: China Aerospace Science and Technology Corporation; CASIC: China Aerospace Science and Industry Corporation.

**Table 2. Description of the disciplinary classification system for statistics.**

| Name of the first-tier discipline | Number of second-tier disciplines | Number of third-tier disciplines |
|---|---|---|
| Philosophy | 1 | 10 |
| Economics | 4 | 34 |
| Law | 6 | 64 |
| Education | 3 | 31 |
| Literature | 3 | 83 |
| History | 1 | 11 |
| Natural Science | 14 | 98 |
| Engineering | 31 | 281 |
| Agriculture | 7 | 60 |
| Medicine | 11 | 58 |
| Military | 10 | 21 |
| Management | 10 | 59 |
| Arts | 5 | 38 |

Based on the collated data, each discipline was classified according to the third-tier classification system and descriptive statistics were retrieved from the first- and second-tier classifications to better understand the structure of the industry's HR needs by discipline.

## 4.2 Discipline structure of the space industry's HR needs

**4.2.1 The discipline structure on a macro scale.** Considering the specific value of each level of academic degree, weights of 1, 2, and 3 were apportioned for the bachelor's, master's, and doctorate degree levels, respectively, to estimate the recruitment numbers of each third-tier discipline. This study treated the weighted quantity of each discipline as an industry human resource needs score (HRNS). Similarly, each university's human resource potential score (HRPS) was calculated. By counting the HRNSs, the proportion of first-tier disciplines was obtained. Among the 13 first-tier disciplines, engineering accounted for the absolute majority (Table 3).

Next, a three-field classification of disciplines was developed, and the proportion of HR needs for each of the three fields (management, literature, economics, and law were combined into a social science category) were calculated (Fig 2).

**Table 3. Proportion of the space industry HR needs by first-tier discipline.**

| Ranking | Discipline category | Percentage |
|---|---|---|
| 1 | Engineering | 91.16% |
| 2 | Management | 3.86% |
| 3 | Natural Science | 2.51% |
| 4 | Literature | 0.79% |
| 5 | Economics | 0.79% |
| 6 | Law | 0.35% |
| 7 | Medicine | 0.15% |
| 8 | Arts | 0.15% |
| 9 | Military | 0.11% |
| 10 | Education | 0.08% |
| 11 | Philosophy | 0.04% |
| 12 | Agriculture | 0.02% |
| 13 | History | 0.01% |

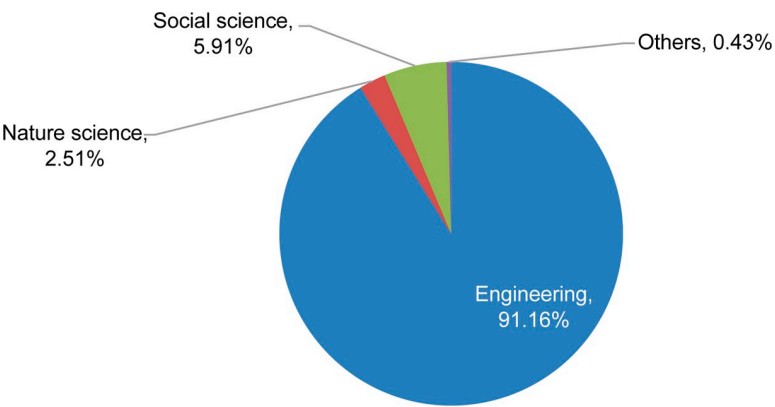

**Fig 2. The proportion of disciplines needed by the space industry classified by engineering, natural science, social science, and others.**

**4.2.2 Discipline structure in the engineering, natural, and social science fields.** By considering HRNSs among second-tier disciplines in the engineering field, the proportion of each discipline was obtained (Fig 3). With the assistance of an expert panel, based on found proportions, the top fifteen disciplines were designated as the primary disciplines, which represented 99.49% of the HRNSs in the engineering field.

Similarly, the proportion of each discipline in the natural science field was obtained (Fig 4). The top six disciplines were identified as the primary disciplines, accounting for 98.44% of the HRNSs in the natural science field.

In the social science field, the top eleven disciplines were identified as the primary disciplines, accounting for 95.58% of the social science field HRNSs (Fig 5).

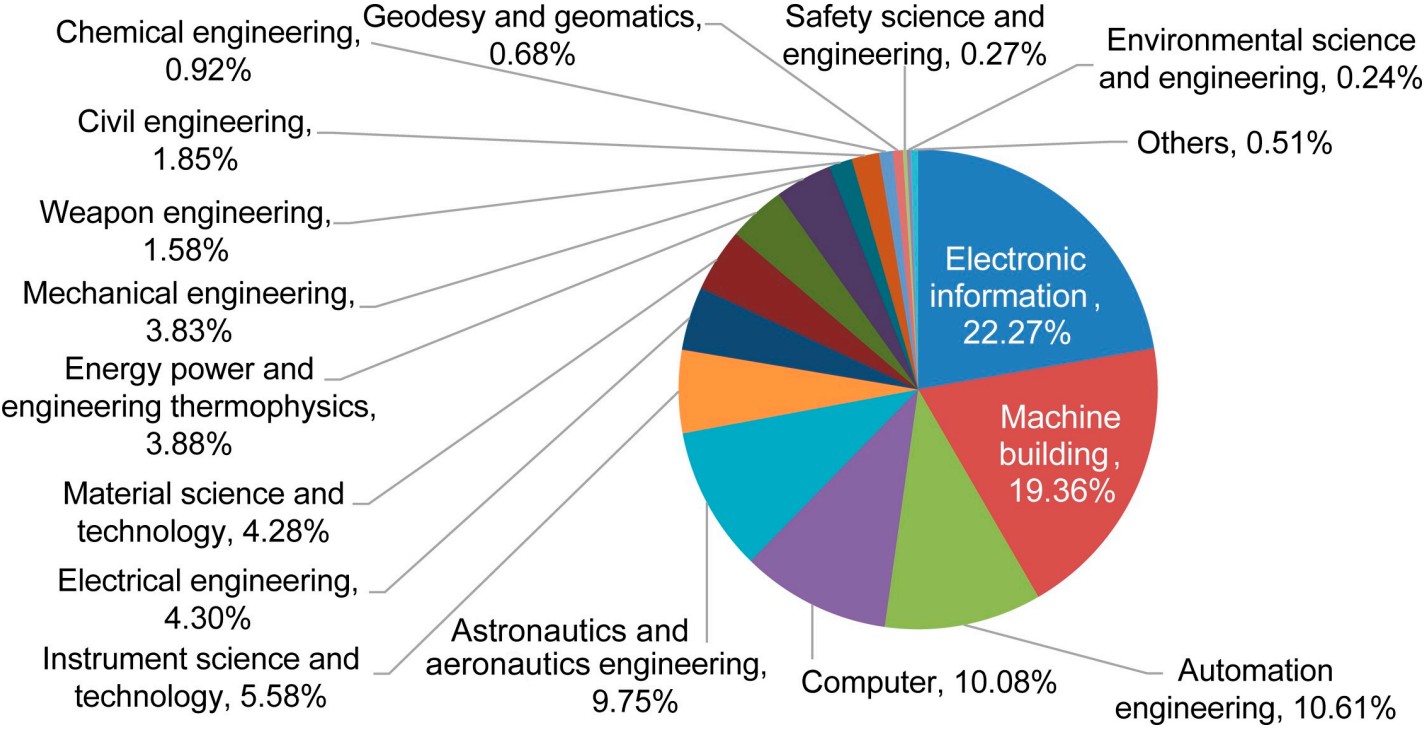

**Fig 3. The proportion of engineering disciplines needed in the space industry.**

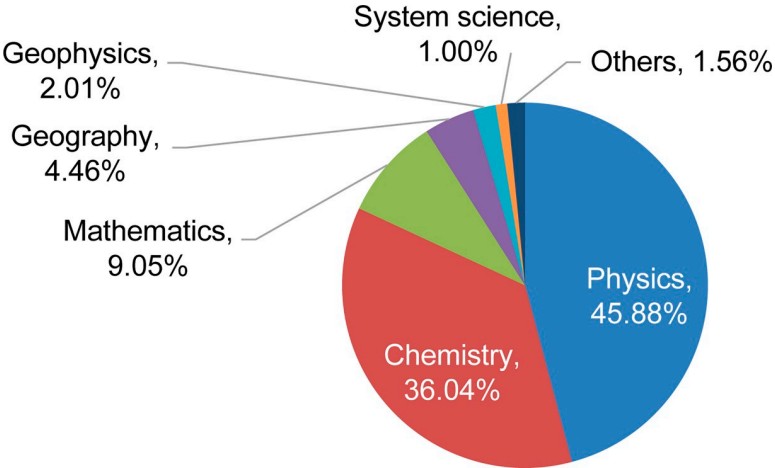

**Fig 4. The proportion of natural science disciplines needed in the space industry.**

## 5. Establishment of the index system

### 5.1 The expert panel

A national panel composed of eight space industry, defense economics, and higher education management experts contributed to the establishment of the space industry supporting

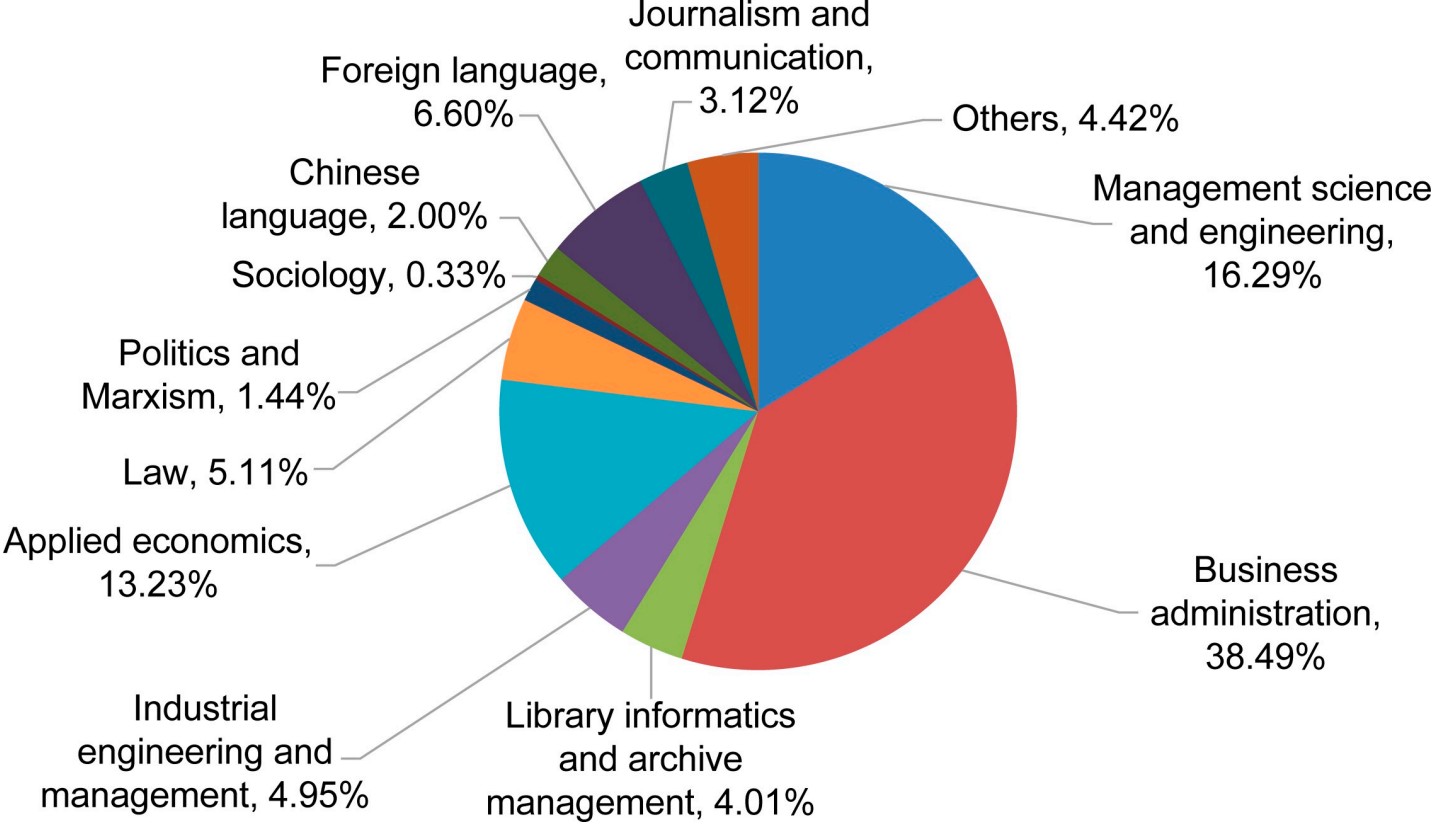

**Fig 5. The proportion of social science disciplines needed in the space industry.**

potential index (SISPI). These experts included senior space industry managers in the CASC, professors in defense economics and management, higher education experts, and senior career advisors, among others. This study selected the indicators and assigned their weights after consultation with the panel.

## 5.2 Selection, definition, and weighting of the indicators

Brainstorming [35, 36], the Delphi method [37], and the analytic hierarchy process (AHP) [38, 39] were employed as the methods of selecting, defining, and weighting the indicators.

First, the research team coordinated and organized online brainstorming sessions to generate the SISPI indictors. Three sessions were held between 21 and 23 June 2019, attended by the eight panelists. Prior to the sessions, all experts were informed of the background, purpose, and research materials of the study. During the first session, the experts discussed the potential scope and criteria of indicators in structured brainstorming. The panelists agreed that the indicators should include the education, research, and management (in the later Delphi process, this was adjusted to culture) domains. All indicators were expected to be quantitative, objective, credible, reasonable, specific, and cost-effective. Indicators should also have readily available data, be easy to explain and understand, and possess long term stability. In the second session, a list of indicators was generated by another structured brainstorming. As a result, 5 Layer-I, 14 Layer-II, and 36 Layer-III indicators mentioned by the experts culminated a long list (some indicators overlap). All the indicators were assigned a brief definition, and all Layer-III indicators possess an assessment strategy. By a reverse brainstorming in the third session, the indicators in the long list were criticized, refined, and merged, to develop a more concise list containing 3 Layer-I, 8 Layer-II, and 24 Layer-III potential indicators.

Second, based on the short list of indicators, a Delphi process using back-to-back e-mail expert consulting from 26 June to 15 July 2019 was conducted. The experts' response rate was 100%. In the first round, each expert on the panel was required to submit their own structure for the index design with a brief explanation. In their designs, each indicator from the list was selected but was not ranked. After summarizing the designs in the first round and sending them to each expert anonymously, the second round of consulting began. In the second round, each expert was asked to revise their design by considering the other experts' designs. This process was repeated over four rounds and achieved consensus; the process ended thereafter. The result revealed a SISPI structure with 3 Layer-I, 7 Layer-II, and 18 Layer-III indicators.

Third, based on the SISPI structure, a questionnaire was developed and an AHP was initiated to generate the weights of the Layer-I and part of the Layer-II indicators (Layer-II indicators for supporting potential in engineering, natural, and social science education were weighted by their proportion of the HRNS). During the AHP, each expert on the panel answered the questionnaire to compare the indicators within each layer in a pairwise manner. The importance levels (scores) were as follows: equal (1), moderate (3), strong (5), very strong (7), and extreme importance (9); the intermediate values between the adjacent scale values were 2, 4, 6, and 8, and the unimportance score was the reciprocal of importance. The weights were calculated by the geometric mean method, and the consistency property of each matrix was confirmed to be acceptable (CR values close to 0). The means (round number) of the experts' results were used as the final weights of the index. Finally, a three-layer index system (Table 4) was established. Table 5 provides the definitions of the Layer III indicators. Because the indicators' value in Layer III was used in the formula, they did not need to be weighted in the AHP.

Table 4. Indicators and their weights in the SISPI system.

| Layer-I | Weights within the total index | Layer-II | Weights within the total index | Weights within layer-II | Layer-III |
|---|---|---|---|---|---|
| Education | 45% | Supporting potential in engineering education | 41.18% | 91.51% | Engineering education direction |
| | | | | | The proportion of related engineering disciplines |
| | | Supporting potential in natural science education | 1.37% | 3.05% | Natural science education direction |
| | | | | | The proportion of related natural science disciplines |
| | | Supporting potential in social science education | 2.45% | 5.45% | Social science education direction |
| | | | | | The proportion of related social science disciplines |
| Research | 35% | Publications | 24.5% | 70% | Internationally published articles |
| | | | | | Nationally published articles |
| | | | | | International conference papers |
| | | | | | Domestic conference papers |
| | | | | | Master's theses |
| | | | | | Doctoral dissertations |
| | | Patents | 10.5% | 30% | Patents |
| Culture | 20% | Strategy | 14% | 70% | Schools and colleges |
| | | | | | A higher-level research organization |
| | | | | | A medium-level research organization |
| | | | | | A lower-level research organization |
| | | Tradition | 6% | 30% | The tradition of space industry-related research |

## 5.3 Indicator assessment and scoring criteria

**5.3.1 Calculation of education indicators.** A cosine similarity measure represents the similarity between two vectors of an inner product space, which is often used to measure similarity in documents for text analysis [40]. Some scholars have used the method in employment analysis to correlate jobs and people [41, 42], to which this study referred. Each university's HRPS was calculated for each discipline by counting a university's enrollment plan for bachelor's, master's, and doctorate programs, which were weighted as 1, 2, and 3, respectively. A similarity assessment was conducted for the engineering, natural science, and social science education fields. To evaluate the similarity between the discipline setting of a university and the HR needs of the space industry, the cosine similarity method was employed [40]. The formula is as follows:

$$\cos\theta = \frac{A \cdot B}{\|A\| \cdot \|B\|} = \frac{\sum_{i=1}^{n} (X_i \cdot Y_i)}{\sqrt{\sum_{i=1}^{n} (X_i)^2} \cdot \sqrt{\sum_{i=1}^{n} (Y_i)^2}},$$

where $X_i Y_i$ is a vector element and $\cos\theta$ reflects the similarity between two vectors, which represents the similarity between the space industry giants' HR needs and a university's discipline setting. The characteristic vectors of each field were developed using the main disciplines as elements and were assigned values according to the disciplines' HRNS or HRPS, for the corporations and universities. Cosine similarity tests for each of the three respective fields were conducted.

The cosine similarity appropriately reflected the discipline setting direction; however, it did not indicate the number of students with related disciplines who are university educated.

**Table 5. Definition of primary indicators (Layer-III).**

| Education | Research | Culture |
|---|---|---|
| Engineering education direction: The similarity between the engineering discipline setting in a university and the HR needs of the space industry, calculated by the cosine similarity method. | Internationally published articles: A university's space industry-related articles published in peer-reviewed English language journals; we included articles published before September 2019. | Colleges and schools: Number of colleges and schools (should be an entity with graduate education function) directly affiliated to a university and related to the space industry. |
| The proportion of engineering-related discipline: Proportion of students in space industry-related engineering disciplines in a university's engineering field, calculated by the number of students with a related engineering discipline divided by the number of all students with an engineering discipline in a university. | Nationally published articles: A university's articles related to the space industry published in peer-reviewed Chinese language journals; we included articles published before September 2019. | Higher-level research organization: Number of space-industry related research organizations recognized by China's central government at a national or international level. |
| Natural science education direction: Similarity between the natural science discipline setting in a university and the HR needs of the space industry, calculated by the cosine similarity method. | International conference papers: A university's space industry-related international conference papers written in English; we included articles published before September 2019. | Mid-level research organization: Number of space industry-related research organizations recognized by a ministry of China's central government or provincial government at a ministerial or provincial level. |
| The proportion of natural science-related disciplines: Proportion of students in engineering space industry-related disciplines in a university's natural science field, calculated by the number of students with a related natural science discipline divided by the number of all students with an engineering discipline in a university. | Domestic conference papers: A university's space industry-related conference papers written in Chinese; we included articles published before September 2019. | Lower-level research organization: Number of space industry-related research organizations not officially recognized by governments higher than the provincial level. |
| Social science education direction: Similarity between the social science discipline setting in a university and the HR needs of the space industry, calculated by the cosine similarity method. | Master's theses: A university's space industry-related master's theses; we included articles published before September 2019. | The tradition of space industry-related research: Cumulative years of space industry-related research of a university, counting the years in which the university published any articles, conference papers, and graduate theses or obtained patents related to the space industry. |
| The proportion of social science-related disciplines: Proportion of students in social science space industry-related disciplines in a university, calculated by the number of students with a related social science discipline divided by the number of all students with a social science discipline in a university. | Doctor's theses: A university's space industry-related doctoral theses; we included articles published before September 2019. | —— |
| —— | Patent: A university's unclassified space-industry related patents published in China; we included patents obtained before September 2019. | —— |

Hence, for adjustment, the HRPS ratio for the main discipline was calculated in each field offered by a university. The cosine similarity value for each field in a university was then combined with their ratios by geometric mean. The scores were normalized to a scale of 0 to 5.

**5.3.2 Calculation of research indicators.** The linear weighting model was used to calculate the publication indicator. For an international article, the journal's five-year impact factor multiplied by 3 was used as a weight. If there was no five-year impact factor, a weight of 3 was assigned. Other publications, that is, domestic articles, domestic conference papers, international conference papers, master's theses, and doctorate theses, were weighted as 1, 0.5, 1.5, 2, and 3.5, respectively.

Patent indicators were calculated by summing a university's number of patents. Typically, if a patent is registered internationally, it is first applied domestically. Thus, this study examined only domestic patents in mainland China.

For data retrieval, this study searched author affiliations and subject keywords related to the space industry to identify the publications and patents of each university in the CNKI database.

The keywords included "astronautics," "space flight," "launch vehicle" or "rocket," "missile," "satellite," "spacecraft," "space station," "anti-missile," "anti-satellite," "lunar exploration," "deep space exploration," and "near-space flight."

**5.3.3 Calculation of culture indicators.** The indicator for strategy (reflected by the number and level of professional space industry-related science and technology organizations) was calculated using the linear weighting model. In the model, colleges and schools and higher-, medium-, and low-level research organizations were weighted as 15, 9, 3, and 1, respectively. If an organization was considered partially related to the space industry by the expert panel, its score was reduced by half.

The indicator for tradition was measured qualitatively based on the number of years a university had been engaged in research. According to the retrieval results for literature, the years in which a university published any space industry-related articles, conference papers, graduate theses, or patents were calculated and treated as representative of tradition.

All the indicators of publication, patents, strategy, and tradition were normalized to a scale of 0 to 5 by the formula $s = 5log_{10}(x)/ log_{10} (x_{max})$, where $s$ is a normalized score, $x$ is its original score, and $x_{max}$ is the maximum score.

**5.3.4 Calculation and further analyses.** After obtaining the three indicator domains, the SISPI was calculated using a linear weighting model according to the weights set in the index system. The SPSS 25 software was employed for further statistics, after obtaining universities' SISPIs. An independent-sample t-test was used to analyze the difference between the seven SEUs and the other universities in the 985 Project. A Pearson correlation test was conducted to analyze the correlation between the indicators. The k-means algorithm was utilized to classify the universities using Layer-II indicators as elements of each university's vector.

# 6. Results

Table 6 shows the scores of SISPI, Layer-I, and Layer-II indicators for the 41 universities. A comprehensive ranking of the universities was obtained from the SISPI scores and their supporting potential for the space industry. Results show that the highest, HIT, scored 4.14, and the lowest, MUC, scored 0.69. The mean SISPI is 2.73 ± 0.82, while its median is 2.72.

The means, standard deviations, and coefficients of variation of SISPI, Layer-I, and Layer-II indicators were obtained from the descriptive statistics (Table 7).

To compare all the indicators for the universities, the scores of Layer I and Layer II were ranked. Table 8 lists the top five universities by indicator. Among the Layer-I indicators, irrespective of the education, research, or culture fields, the top five are former MMB-affiliated universities. Among the Layer-II indicators, in the engineering field, the top five are former MMB-affiliated universities. However, none of the top five in the natural and social science fields are former MMB-affiliated universities. For the fields of publication, patents, and strategy, four of the top five are former MMB-affiliated universities. The top five in the field of tradition were combined with the former MMB- and 985 project-affiliated groups.

Expert consultants concluded that if the SISPI score of a university is lower than 1.5, it has weak potential to support the industry (almost incapable of supporting the industry in the main field). If the SISPI score is between 1.5 and 3, the university has medium potential (could partially support the industry). If the SISPI score is higher than 3, the university has a high potential (could support the industry adequately; Fig 6). This standard could also be adapted for the domains of education, research, and culture.

The t-test results indicate that the former MMB-affiliated universities score significantly higher than the 985 Project-listed universities regarding the SISPI and the fields of education, research, and culture at a significance level of 0.05 (two-tailed; Table 9). There are significant

**Table 6. Scores of SISPI, Layer-I, and Layer-II indicators (ranking by SISPI).**

| University | Class* | Engineering | Natural science | Social science | Publications | Patents | Strategy | Tradition | Education | Research | Culture | SISPI |
|---|---|---|---|---|---|---|---|---|---|---|---|---|
| HIT | MMB, 985 | 3.33 | 2.22 | 4.73 | 4.65 | 4.77 | 5.00 | 4.86 | 3.36 | 4.69 | 4.95 | 4.14 |
| NUAA | MMB, 211 | 3.36 | 3.85 | 4.55 | 4.64 | 4.27 | 4.12 | 4.39 | 3.43 | 4.53 | 4.22 | 3.97 |
| NWPU | MMB, 985 | 3.18 | 3.74 | 4.30 | 4.45 | 4.53 | 4.57 | 5.00 | 3.25 | 4.47 | 4.72 | 3.97 |
| BIT | MMB, 985 | 3.21 | 3.40 | 4.75 | 4.20 | 4.28 | 3.87 | 4.67 | 3.30 | 4.22 | 4.15 | 3.79 |
| HEU | MMB, 211 | 3.38 | 2.59 | 4.56 | 4.30 | 3.39 | 3.75 | 4.28 | 3.42 | 4.03 | 3.94 | 3.73 |
| BUAA | MMB, 985 | 2.50 | 2.90 | 4.40 | 4.34 | 5.00 | 4.71 | 4.67 | 2.62 | 4.54 | 4.70 | 3.71 |
| NUST | MMB, 211 | 3.50 | 3.89 | 4.45 | 4.62 | 3.80 | 1.71 | 4.44 | 3.55 | 4.38 | 2.67 | 3.66 |
| UET | 985 | 2.94 | 3.80 | 4.51 | 4.41 | 3.54 | 3.08 | 4.22 | 3.04 | 4.15 | 3.48 | 3.52 |
| THU | 985 | 2.74 | 3.81 | 3.00 | 4.01 | 4.34 | 3.62 | 4.90 | 2.78 | 4.11 | 4.06 | 3.50 |
| HUST | 985 | 2.95 | 4.75 | 4.80 | 3.97 | 2.88 | 3.29 | 3.96 | 3.12 | 3.64 | 3.53 | 3.38 |
| SJTU | 985 | 2.53 | 2.21 | 4.83 | 4.09 | 3.55 | 3.46 | 4.82 | 2.68 | 3.93 | 3.94 | 3.37 |
| ZJU | 985 | 2.59 | 4.11 | 3.91 | 4.02 | 3.97 | 3.08 | 4.49 | 2.73 | 4.01 | 3.57 | 3.34 |
| WHU | 985 | 1.58 | 4.10 | 4.59 | 4.34 | 4.18 | 3.98 | 4.39 | 1.87 | 4.29 | 4.12 | 3.16 |
| DLUT | 985 | 2.50 | 4.59 | 4.89 | 3.83 | 2.98 | 3.08 | 4.10 | 2.70 | 3.57 | 3.44 | 3.15 |
| XJTU | 985 | 2.43 | 2.29 | 4.84 | 3.44 | 2.88 | 3.08 | 4.49 | 2.57 | 3.27 | 3.57 | 3.01 |
| USTC | 985 | 1.91 | 4.76 | 4.82 | 3.67 | 2.32 | 4.08 | 4.53 | 2.18 | 3.27 | 4.24 | 2.97 |
| TJU | 985 | 2.97 | 3.81 | 4.76 | 3.71 | 3.01 | 0.00 | 4.10 | 3.09 | 3.50 | 1.43 | 2.90 |
| SDU | 985 | 2.45 | 4.76 | 4.69 | 3.51 | 2.48 | 2.50 | 3.10 | 2.68 | 3.20 | 2.71 | 2.87 |
| SEU | 985 | 2.80 | 2.33 | 4.92 | 3.48 | 4.00 | 0.00 | 4.16 | 2.89 | 3.63 | 1.46 | 2.87 |
| CQU | 985 | 2.15 | 1.01 | 4.87 | 3.53 | 2.88 | 3.08 | 3.55 | 2.30 | 3.33 | 3.25 | 2.85 |
| CSU | 985 | 1.94 | 2.28 | 4.89 | 3.31 | 2.75 | 3.19 | 3.65 | 2.12 | 3.14 | 3.35 | 2.72 |
| NJU | 985 | 1.35 | 4.64 | 4.54 | 3.70 | 2.79 | 3.08 | 4.49 | 1.72 | 3.43 | 3.57 | 2.69 |
| PKU | 985 | 1.09 | 4.07 | 4.58 | 3.46 | 3.23 | 3.38 | 4.71 | 1.56 | 3.39 | 3.84 | 2.66 |
| TOU | 985 | 1.86 | 3.22 | 4.82 | 3.07 | 2.59 | 3.08 | 4.22 | 2.08 | 2.92 | 3.48 | 2.65 |
| XMU | 985 | 2.26 | 4.34 | 4.27 | 3.02 | 2.99 | 3.08 | 3.45 | 2.06 | 3.01 | 3.21 | 2.62 |
| SCUT | 985 | 2.50 | 3.56 | 4.84 | 3.37 | 2.77 | 0.00 | 3.81 | 2.67 | 3.19 | 1.34 | 2.59 |
| NEU | 985 | 2.56 | 4.15 | 4.79 | 3.48 | 1.75 | 0.46 | 3.55 | 2.73 | 2.96 | 1.54 | 2.58 |
| JLU | 985 | 2.16 | 4.74 | 4.57 | 3.86 | 2.51 | 0.00 | 3.81 | 2.42 | 3.46 | 1.34 | 2.57 |
| ECNU | 985 | 1.24 | 4.57 | 3.63 | 3.48 | 2.81 | 1.25 | 3.89 | 1.60 | 3.28 | 2.17 | 2.30 |
| HNU | 985 | 2.16 | 1.54 | 4.09 | 3.41 | 1.23 | 0.00 | 3.35 | 2.27 | 2.75 | 1.17 | 2.22 |
| FDU | 985 | 1.03 | 4.49 | 4.60 | 2.71 | 1.36 | 3.08 | 4.10 | 1.54 | 2.31 | 3.44 | 2.19 |
| OUC | 985 | 1.04 | 1.14 | 4.62 | 4.01 | 2.69 | 0.00 | 3.81 | 1.34 | 3.62 | 1.34 | 2.14 |
| SYSU | 985 | 1.34 | 4.32 | 4.60 | 2.09 | 2.20 | 3.08 | 3.10 | 1.70 | 2.12 | 3.09 | 2.13 |
| SCU | 985 | 1.71 | 4.53 | 4.76 | 2.76 | 2.01 | 0.00 | 3.45 | 2.05 | 2.54 | 1.21 | 2.05 |
| LZU | 985 | 1.19 | 4.26 | 4.44 | 3.61 | 0.00 | 0.00 | 3.55 | 1.57 | 2.53 | 1.24 | 1.84 |
| BNU | 985 | 0.20 | 3.97 | 4.15 | 3.19 | 2.20 | 0.00 | 4.16 | 0.71 | 2.89 | 1.46 | 1.62 |
| CAU | 985 | 0.74 | 0.91 | 3.77 | 2.82 | 1.89 | 0.00 | 3.23 | 1.07 | 2.54 | 1.13 | 1.60 |
| NKU | 985 | 1.31 | 4.36 | 2.88 | 2.16 | 1.06 | 0.00 | 3.23 | 1.59 | 1.83 | 1.13 | 1.58 |
| NAFU | 985 | 1.06 | 1.16 | 3.44 | 2.47 | 0.84 | 0.00 | 2.17 | 1.27 | 1.98 | 0.76 | 1.42 |
| RUC | 985 | 0.37 | 2.26 | 4.39 | 2.00 | 0.00 | 0.00 | 3.45 | 0.82 | 1.40 | 1.21 | 1.10 |
| MUC | 985 | 0.63 | 1.24 | 3.30 | 0.98 | 0.00 | 0.00 | 0.00 | 1.00 | 0.69 | 0.00 | 0.69 |

BIT: Beijing Institute of Technology; BNU: Beijing Normal University; BUAA: Beijing University of Aeronautics and Astronautics; CAU: China Agricultural University; CQU: Chongqing University; CSU: Central South University; DLUT: Dalian University of Technology; ECNU: East China Normal University; FDU: Fudan University; HEU: Harbin Engineering University; HIT: Harbin Institute of Technology; HNU: Hunan University; HUST: Huazhong University of Science and Technology; JLU: Jilin University; LZU: Lanzhou University; MUC: Minzu University of China; NAFU: Northwest A&F University; NEU: Northeastern University; NJU: Nanjing University; NKU: Nankai University; NUAA: Nanjing University of Aeronautics and Astronautics; NUST: Nanjing University of Science and Technology; NWPU: Northwestern Polytechnical University; OUC: Ocean University of China; PKU: Peking University; RUC: Renmin University of China; SCU: Sichuan University; SCUT: South China University of Technology; SDU: Shandong University; SEU: Southeast University; SJTU: Shanghai Jiao Tong University; SYSU: Sun Yat-sen University; THU: Tsinghua University; TJU: Tianjin University; TOU: Tongji University; UET: University of Electronic Science and Technology; USTC: University of Science and Technology of China; WHU: Wuhan University; XJTU: Xi'an Jiaotong University; XMU: Xiamen University; ZJU: Zhejiang University.

*MMB: former Ministry of Machine Building industry affiliated university; 985: universities listed in the 985 Project; 211: universities listed in the 211 Project.

**Table 7. Descriptive statistics of all indicators and their coefficients of variation.**

| Indicator | Layer | Mean | SD* | CV** |
|---|---|---|---|---|
| SISPI | —— | 2.73 | 0.82 | 0.30 |
| Education | I | 2.28 | 0.78 | 0.34 |
| Research | I | 3.29 | 0.89 | 0.27 |
| Culture | I | 2.76 | 1.33 | 0.48 |
| Engineering | II | 2.07 | 0.90 | 0.43 |
| Natural science | II | 3.38 | 1.22 | 0.36 |
| Social science | II | 4.42 | 0.52 | 0.12 |
| Publications | II | 3.52 | 0.80 | 0.23 |
| Patents | II | 2.75 | 1.27 | 0.46 |
| Strategy | II | 2.14 | 1.75 | 0.82 |
| Tradition | II | 3.91 | 0.87 | 0.22 |

SD: standard deviation; CV: coefficient of variation.

differences between the two groups in publication, patents, strategy, tradition, and engineering regarding Layer-II indicators at the same significance level. In the cases where equal variance was not assumed, t'-test was used to adjust the results by SPSS automatically.

The Pearson test results show that three Layer-I indicators are highly positively correlated with each other at a significance level of 0.05 (two-tailed), for which the correlation coefficient ($r$) of education and research is 0.76 (P<0.01), that of education and culture is 0.54 (P<0.01), and that of culture and research is 0.72 (P<0.01).

**Table 8. Top five universities for Layer-I and Layer-II indicators.**

| Indicators | Layer | Ranking | | | | |
|---|---|---|---|---|---|---|
| | | 1 | 2 | 3 | 4 | 5 |
| Education | I | NUST | NUAA | HEU | HIT | BIT |
| Research | I | HIT | BUAA | NUAA | NWPU | NUST |
| Culture | I | HIT | NWPU | BUAA | USTC | NUAA |
| Engineering | II | NUST | HEU | NUAA | HIT | BIT |
| Natural science | II | USTC | SDU | HUST | JLU | NJU |
| Social science | II | SEU | DLUT | CSU | CQU | XJTU |
| Publications | II | HIT | NUAA | NUST | NWPU | UET |
| Patents | II | BUAA | HIT | NWPU | THU | BIT |
| Strategy | II | HIT | BUAA | NWPU | NUAA | USTC |
| Tradition | II | NWPU | THU | HIT | SJTU | PKU |

BIT: Beijing Institute of Technology; BNU: Beijing Normal University; BUAA: Beijing University of Aeronautics and Astronautics; CAU: China Agricultural University; CQU: Chongqing University; CSU: Central South University; DLUT: Dalian University of Technology; ECNU: East China Normal University; FDU: Fudan University; HEU: Harbin Engineering University; HIT: Harbin Institute of Technology; HNU: Hunan University; HUST: Huazhong University of Science and Technology; JLU: Jilin University; LZU: Lanzhou University; MUC: Minzu University of China; NAFU: Northwest A&F University; NEU: Northeastern University; NJU: Nanjing University; NKU: Nankai University; NUAA: Nanjing University of Aeronautics and Astronautics; NUST: Nanjing University of Science and Technology; NWPU: Northwestern Polytechnical University; OUC: Ocean University of China; PKU: Peking University; RUC: Renmin University of China; SCU: Sichuan University; SCUT: South China University of Technology; SDU: Shandong University; SEU: Southeast University; SJTU: Shanghai Jiao Tong University; SYSU: Sun Yat-sen University; THU: Tsinghua University; TJU: Tianjin University; TOU: Tongji University; UET: University of Electronic Science and Technology; USTC: University of Science and Technology of China; WHU: Wuhan University; XJTU: Xi'an Jiaotong University; XMU: Xiamen University; ZJU: Zhejiang University.

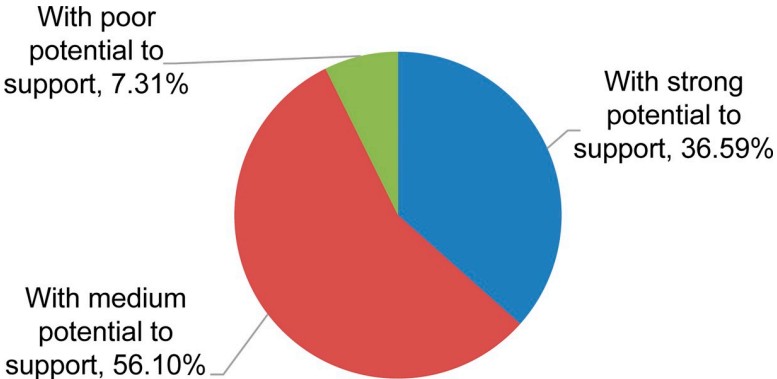

**Fig 6. Different levels of supporting potential among the universities.**

The correlations between Layer-II indicators were also tested. The results show that engineering education is highly positively correlated with publication ($r = 0.73$, P<0.01) and patents ($r = 0.71$, P<0.01) and moderately positively correlated with tradition ($r = 0.50$, P<0.01), strategy ($r = 0.49$, P<0.05), and social science education ($r = 0.35$, P<0.05). Natural science education is moderately positively correlated with tradition ($r = 0.33$, P<0.05). Social science education is moderately positively correlated with publications ($r = 0.39$, P<0.05) and tradition ($r = 0.40$, P<0.05). Publication is highly positively correlated with patents ($r = 0.80$, P<0.01) and tradition ($r = 0.74$, P<0.01) and moderately positively correlated with strategy ($r = 0.54$ P<0.01). Strategy is moderately positively correlated with tradition ($r = 0.57$, P<0.01).

For the k-means algorithm, the number of cluster centers was set from three to five; finally, four cluster settings were considered an appropriate setting to classify the universities (Table 10). The results show the convergence achieved within the four iterations.

SDU, NEU, JLU, ECNU, FDU, SYSU, SCU, LZU, BNU, and NKU are classified into cluster 2. TOU, SEU, SCUT, HNU, and OUC are classified into cluster 3. CAU, NAFU, RUC, and MUC are classified into cluster 4. The other 22 universities are classified into cluster 1, which includes the seven former MMB-affiliated universities.

**Table 9. Means and standard deviations of the SISPI, Layer-I, and Layer-II indicators in groups of former MMB-affiliated universities and 985 project-listed universities.**

| Groups | Layer | Former MMB-affiliated universities | 985 project-affiliated only universities |
|---|---|---|---|
| SISPI | ——— | 3.85±0.18** | 2.50±0.70 |
| Education | I | 3.28±0.31** | 2.07±0.68 |
| Research | I | 4.41±0.22** | 3.06±0.80 |
| Culture | I | 4.19±0.76** | 2.47±1.24 |
| Publications | II | 4.46±0.18** | 3.32±0.74 |
| Patents | II | 4.29±0.55* | 2.43±1.14 |
| Strategy | II | 3.96±1.09** | 1.77±1.62 |
| Tradition | II | 4.62±0.26* | 3.76±0.88 |
| Engineering | II | 3.21±0.33** | 1.83±0.79 |
| Natural science | II | 3.23±0.66 | 3.41±1.31 |
| Social science | II | 4.53±0.17 | 4.39±0.57 |

*Notes.*

\* P<0.05 (by t-test)

\*\* P<0.05 (by t'-test) versus only 985 Projects-listed universities.

**Table 10. Four cluster centers of the k-means algorithm by Layer-II indicators.**

| Layer-II indicator | Cluster 1 | Cluster 2 | Cluster 3 | Cluster 4 |
|---|---|---|---|---|
| Engineering | 2.51 | 1.52 | 2.29 | 0.70 |
| Natural science | 3.48 | 4.42 | 2.48 | 1.39 |
| Social science | 4.54 | 4.31 | 4.65 | 3.73 |
| Publications | 3.96 | 3.09 | 3.60 | 2.07 |
| Patents | 3.54 | 1.84 | 2.74 | 0.68 |
| Strategy | 3.52 | 1.04 | 0.00 | 0.00 |
| Tradition | 4.38 | 3.59 | 3.85 | 2.21 |

## 7. Discussion

### 7.1 Main findings and interpretation

China's space industry engineering disciplines account for an overwhelming proportion of its HR needs. It would be to meet their core mission's needs of China's space industry in engineering research and development and volume production of space, missile, and other defense systems, which is in accordance with their official declaration, "empowering the army with science and technology and serving the nation with aerospace technology" [29, 30]. The engineering field requires personnel with knowledge of electronics, machine building, automation engineering, computers, and space engineering. In contrast, the natural science field requires physics and chemistry graduates, but the level of need is minimal. It is interesting that the HR needs in social science are twice as high as those in natural science. This could be explained by the industry's need for management personnel with an educational background in business administration, management, and economics to enhance their administrative capacity. If educational authorities or university leadership commits to supplying HR to the space industry, they may enhance their discipline construction according to the fields. Meanwhile, corporation executives should give more attention to universities with the desired discipline characteristics when recruiting on campuses.

The three domains of education, research, and culture have a mutual synergistic effect. However, due to the nature of the correlation tests conducted in this study, it is difficult to confirm the existence of a causal relationship. In the long term, a university's culture (strategy at the management level and tradition at the faculty level) can affect its education and research output and functions. If a university is dominated by the management-and faculty-level culture, its support potential for the industry could be significant; by contrast, without that culture, it could not have such potential.

According to the coefficients of variation, the difference in university development strategies for space industry-related fields is higher than other indicators. This difference influences university respect toward providing support to the space industry, placing universities on a different development path. Diversity is also evident in university patents, publications, and engineering education. The implication is that the direction of engineering education is quite different among the universities for supporting the space industry, whereas the capability gap among universities regarding patents and publications in space industry-related fields is significant. As a highly weighted aspect of SISPI, engineering education is crucial for a university to support the space industry. Meanwhile, social science education is not such a critical diversity factor for supporting the space industry, which means that the HR potential for the space industry with respect to social science is interchangeable among universities.

Within the education domain, the relationships between the three fields were not strong, implying that different education fields minimally influence each other in their supporting

potential for the space industry. In other words, at the macro-level of education, the development of major categories of disciplines related to the space industry is relatively independent. Within the research domain, the high correlation between publications and patents suggests that theoretical achievements can be converted to practical achievements in space industry-related fields. Furthermore, within the culture domain, strategy and tradition were highly correlated, which implies that strategy generated by the leadership and faculty practices is mutually beneficial.

Results further indicated that engineering education direction and research achievements could have deep mutual involvement. Research output related to the space industry is rooted in a solid disciplinary foundation that is coordinated with the industry. On the contrary, the direction of natural and social science education has less effect on a university's space industry-related research output as well as weaker relationships with space industry-related strategy and tradition. This finding implies that a university, with or without a culture that is conducive to the space industry, may provide support to the space industry in natural and social science education.

Overall, the results show favorable conditions for China's university system to support the development of its space industry. However, the average potential of all of China's high-quality universities is inferior to that of the seven SEUs. More than one-third of the elite universities have satisfactory potential to support the space industry, and more than half of them could be defined as having certain support potential; a few high-quality universities have minimal potential to support the space industry. Meanwhile, the seven former MMB-affiliated SEUs, irrespective of whether they are listed in the 985 Project, have extraordinary advantages in supporting the space industry from an overall perspective as well as in specific aspects such as engineering education, research output, and culture of fostering space industry-related education and research. There were fewer differences among the seven SEUs from an overall perspective; however, specifically, each SEU has its own merits.

According to the cluster analysis, the universities could be classified into four groups:

Group I (Cluster 1: Strategic partner) is characterized by a more explicit strategy and a longer tradition in space industry-related education or research. This group meets the majority of the industry's HR needs in each primary field of education. Moreover, it produces fruitful theoretical and practical research output.

Group II (Cluster 2: Armchair strategist) has a strategy for space industry-related education or research, although it is relatively weak. The group's faculties focus on industry-related research in the long term and have exhibited theoretical achievements. However, practical research output is minimal. Regarding education, this group's natural and social science education could meet the HR needs of the space industry; however, engineering education is weak in this aspect.

Group III (Cluster 3: Potential friend) contains faculty that is interested in long-term space industry-related research. Moreover, this group has recorded many theoretical and some practical industry-related achievements. The leadership, however, has no explicit strategy focus on space industry-related education or research. The disciplinary system could meet the industry's HR needs for social science and partially meet the needs for engineering and natural science.

The strategy of Group IV (Cluster 4: Outsider) is not focused on space industry-related fields, and the faculty is not particularly interested in these fields either. Thus, research output in the related fields is less. This group's education could meet the space industry's HR needs only in the social science field.

Based on this classification, members of Group I could be defined as space industry-friendly universities. For the industry authority, it is important to deepen cooperative relations with

industry-friendly universities, which could enrich their internal and external intellectual capital and enhance their innovation capability and performance. The industry should also give attention to the universities from Groups II and III to broaden their intellectual capital base. For example, motivating Group II universities' leadership and faculties, allowing them to accomplish more cooperative research practices with the space industry; conducting more public relations activities in campus recruitment in Group III, attracting graduates with disciplines that the space industry needs.

## 7.2 Strengths and limitations

This study is the first to establish and implement an index system to assess China's universities' capability to support the country's space industry development. The objective indicators in the index system can avoid excessive biases due to over-dependency on subjective judgments from experts or managers. Public data sources were obtained, and all the indicators were calculated based on the data. The indicators selected were compact and easy to access; thus, the process was low cost, and managers can use the indicators in their administrations. Although a few universities included in this study could be considered social science-oriented, they are comparable in SISPI scores. In this study, scores of social science-oriented universities, such as RUC, in most aspects were clearly lower than those of the SEUs. The results indicated that SISPI scores can clearly differentiate between these types of universities.

This study has several limitations. The primary limitation is the limited data sources due to the low publicity of both universities and the space industry. Consequently, some important data could not be obtained; thus, some valuable indicators could not be established. This study included articles, papers, theses, and patents that can be openly accessed, while the classified literature was not included. This exclusion of the classified literature could limit the study; the unclassified research literature reflects the research achievement in space industry-related fields only to some extent. In addition, to reduce the cost of the study and avoid loss of sample size due to refusal to respond by participants, the personnel number, and the size of investments by professional organizations were not considered. Instead, only the number and level of organizations were considered, which could have caused some inaccuracies.

Furthermore, university tradition concerning space industry-related research was represented by the accumulated years of publications, which could cause scaling error. As China's journal system was gradually rebuilt after the "reopening" policy (after December 1978), the online availability of publications and patents began only after the 1990s. However, China's space industry was founded in the late 1950s, and, at that time, some universities were already involved in the industry. Thus, the tradition indicator offers a compromised selection to assess a university's related research activity tradition after the "Opening of China."

Moreover, SISPI cannot evaluate graduates' desire to serve the industry. In China, however, universities are encouraged to supply HR to state-run defense sectors, such as the space industry, as can be seen in some universities' annual employment quality report, which is required by the authority [43]. It is apparent that in these reports, universities emphasize graduate student employment in the defense and space industries [44–47]. Although graduates would not automatically work for the space industry, this study evaluates the supporting potential of each university for the authority. Under certain conditions, such as possible industrial mobilization, the authority could provide guidance for employment according to the evaluation.

## 8. Conclusion

This study created the SISPI by conducting an empirical study with elite universities in China. From a macro perspective, China's university system can reliably provide intellectual capital to

the space industry. However, the supporting potential is unbalanced; it is concentrated in some universities, such as the former MMB-affiliated universities on which the industry depends. From a specific perspective, the HR needs of China's space industry are primarily confined to engineering fields, which can be adequately supplied by high-quality universities depending upon their type. Although the seven SEUs were officially delinked from the industry for approximately 20 years and developed under state plans with other high-quality universities, they still have an advantage in most fields in supporting the space industry.

Over the past several decades, China has witnessed rapid and observable economic progress. Space activities have accelerated in recent years [48]. If China continues to invest in boosting its space industry and fulfilling its space ambitions, state resources, including the university system, would be mobilized. Thus, the potential of the system and each university should be quantitatively ascertained through management tools such as the SISPI. This study provides evidence that the SISPI is a feasible, simple, and practical tool for evaluating the extent to which a university could support the space industry. The SISPI can help the industry's leadership to identify space industry-friendly universities for HR recruitment and R&D collaboration. The SISPI can also guide a university in committing to supporting the space industry, evaluate its potential, and ascertain the gaps between its capabilities and the industry's needs.

The SISPI, however, cannot be directly generalized to other countries due to the HRNSs in this study being customized to China's space industry; nevertheless, the indicators, weights, and HR supply-need matching method could be considered a valuable reference to related worldwide research, especially for the current or emerging great military powers such as the US, Russia, and India. If the target country's space industry HRNSs can be obtained, it is easy to customize the SISPI in accordance with these circumstances. Moreover, a similar method could be generalized to develop indexes for other defense industry sectors, such as aviation, shipbuilding, nuclear, and ordnance, to observe China.

## Acknowledgments

The useful comments and constructive suggestions by anonymous referees of the space industry are gratefully acknowledged.

## Author Contributions

**Conceptualization:** Xiaoxiao Li.

**Data curation:** Wei Niu.

**Formal analysis:** Wei Niu.

**Funding acquisition:** Xiaoxiao Li.

**Investigation:** Xiaoxiao Li.

**Methodology:** Xiaoxiao Li.

**Project administration:** Xiaoxiao Li.

**Resources:** Wei Niu.

**Software:** Wei Niu.

**Supervision:** Xiaoxiao Li.

**Validation:** Xiaoxiao Li.

**Visualization:** Xiaoxiao Li.

**Writing – original draft:** Xiaoxiao Li.

**Writing – review & editing:** Xiaoxiao Li.

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
