## [Decision Letter · Decision Letter 0]

14 Oct 2020

PONE-D-20-25196

Support potential of elite civilian universities for China’s space industry: higher educational mobilization capacity for China’s space ambition

PLOS ONE

Dear Dr. Li,

Thank you for submitting your manuscript to PLOS ONE. After careful consideration, we feel that it has merit but does not fully meet PLOS ONE’s publication criteria as it currently stands. Therefore, we invite you to submit a revised version of the manuscript that addresses the points raised during the review process.

We look forward to receiving your revised manuscript.

Kind regards,

Bing Xue, Ph.D.

Academic Editor

PLOS ONE

Journal Requirements:

2.We note that you have stated that you will provide repository information for your data at acceptance. Should your manuscript be accepted for publication, we will hold it until you provide the relevant accession numbers or DOIs necessary to access your data. If you wish to make changes to your Data Availability statement, please describe these changes in your cover letter and we will update your Data Availability statement to reflect the information you provide.

Reviewers' comments:

Reviewer's Responses to Questions

**Comments to the Author**

1. Is the manuscript technically sound, and do the data support the conclusions?

Reviewer #1: Yes

2. Has the statistical analysis been performed appropriately and rigorously? 

Reviewer #1: Yes

3. Have the authors made all data underlying the findings in their manuscript fully available?

Reviewer #1: No

4. Is the manuscript presented in an intelligible fashion and written in standard English?

Reviewer #1: Yes

5. Review Comments to the Author

Reviewer #1: In this article, the authors presented us with the human resources situation of China’s major space industry companies and provided a tool, which is called the space industry supporting potential index (SISPI), to measure the ability of higher education institutions supporting space companies. Based on this new technology, the researchers measured 41 universities in China and using K-means clustering method, the researchers obtained four categories of university groupings which showed the obvious heterogeneity of universities in the contribution potential of the aerospace industry.

There is no doubt that this study is a very meaningful and valuable study, it gives us more information about china’ space industry and It motivates us to use an innovative tool to measure the ability of universities to support specific industries.

But it would be better if author could do some modification which is, of course, in my personal opinion.

Major Issue

1. Can you give a clearer concept to civilian universities. I studied this article carefully, but for the meaning of civil universities, especially the connection and difference between civil universities and elite universities, I could not understand. Perhaps, through defining the above two concepts, you will make the article clearer and better.

2. You have collected such fascinating data since 2014 (Line 181), it would be a good choice to do more job description or any other analysis. In particular, can you provide a more detailed table of basic information about collection time, job posting time, job category, discipline, degree requirements, etc. These information are very important for judging whether the sample can be used to represent the population.

3. It made me very confused when I read the table 1 (Line 204). What is your basis in the process of classifying and describing disciplinary classification system. I tried to match the latest version of the Catalogue of undergraduate major of general institution of higher learning with your table, but it didn’t match. So I think it would be better to give more directly information to help reader understand.

4. It is my concern with you assertion that “ China’s space industry engineering disciplines account for an overwhelming proportion of its HR needs…scientific research accounts for significantly less activity.” (Line 475-478). I want to know if you have more evidence to say this. According to my knowledge, the sample companies are large-scale scientific research and production consortium. It is understandable that related companies recruit a large number of engineering talents to manufacture related products. At the same time, engineering talents also have scientific research capabilities and have often participated in innovative scientific research projects. More importantly, in this article, I did not find more evidence that related companies have less scientific research activities. Therefore, it may be better to provide more evidence to support this argument.

Minor Issue

1. It would be better for reader to understand the relevant background and enjoy article by using more ink to introduce the sample companies, especially the personnel scale, personnel structure, and future human resource vision. The above information could be gotten from the website of sample companies. It would be better to put these information to “3. Material and methods” (Line 165) .

2. It would be some mistake in table 6 (Lines 400). The Intersection of Engineering Row and CV should be 0.43 according your data on the table.

3. It would be better to tell us whether you have done a test for homogeneity of variance in table 8 (Lines 445). It would be great to show in the table.

4. It would be better to give a name for the 4 cluster of universities (Line 468 or 536). I believe it would be better to put the name to Line 468 if you can find a good name.

5. Of course, it would make reader more enjoyable by polishing the language and article. For example, you should adjust the table 5 (Line 378) and do more reading to find the mini mistake.

6. PLOS authors have the option to publish the peer review history of their article (what does this mean?). If published, this will include your full peer review and any attached files.

Reviewer #1: No

---

## [Author Response · Author response to Decision Letter 0]

30 Oct 2020

October 30, 2020

PLoS ONE

Dear Editor and Reviewer: 

We wish to thank you for the time and effort you have spent reviewing our manuscript entitled “Support potential of elite civil universities for China’s space industry: higher educational mobilization capacity for China’s space ambition” (PONE-D-20-25196). We are pleased to note that you have found our research work interesting and also pointed out some problems to help us improve the quality of our work. We have studied the comments carefully and have made corrections that we hope meet with approval. The revised sections are marked in the manuscript. The main corrections to the paper and our responses to the reviewer’s comments are as follows:

To Major Issue 1:

Firstly, thank you very much for your valuable comment. 

According to suggestion form a native English language expert, it would be more appropriate perhaps to use “civil service university” instead of “civilian university”. However, the word “civil service university” also is not fully in line with the concept in our study. Thus, we try to use the word “civil university” instead.

We have added a brief definition into the “Materials and methods” section (lines 174-177). In this study, we used the word “civil university” to distinguish between universities that are affiliated with the People’s Liberation Army (PLA), People's Armed Police, Police, or other armed forces. We wished to assess the mobilization potential of the civil branch of the national educational system and identify the non-military universities that have space/missile industry human resources (HR) and research and design (R&D) support capability. Thus, some military universities, such as the National University of Defense Technology of PLA, which are listed in the 985 and 211 Project, were excluded. 

For former Ministry of Machine Building (MMB) universities, it has been mentioned in the “Introduction” (lines 87-89) that “Those seven SEUs are currently affiliated with the Ministry of Industry and Information Technology (MIIT) under the central government”. MIIT is a civil branch of central government; thus, they can be considered as civil universities.

Addressing your concern, I have tried to demonstrate the connection and difference between civil universities and elite universities with a figure (Appendix Fig. 1). However, considering the length of the manuscript, it has not been added to the main text.

To Major Issue 2:

We have added a brief table under the “Materials and methods” section to further describe the data. A more detailed table was added into the appendix (Table 1) for your reference. However, I believe that it is too large to be added to the main text. 

To observe the activities of defense industry corporations in China, the economic cycle was set to more than five years, because the state-run corporations’ development plans are highly dependent on the Five-Year Plan of the state. It is possible to find long tendencies by observing the yearly activities of items such as human resourcing; however, to do more comparison between each year may not deliver very useful information. 

Furthermore, I believe that to describe all the details (collection time, job posting time, job category, discipline, and degree requirements, etc.) in one table is somewhat overwhelming. In most cases, the data contained the jobs’ title; however, this was not used for analysis in this study. Moreover, there were thousands of job titles without standardization in the raw data: classifying the job title is somewhat more difficult than classifying the discipline. I believe that it would be more beneficial to analyze these data in another study in the future. 

CASC and CASIC publish their official HR needs online annually; however, they routinely delete or substitute previous years’ information. Consequently, there is no comprehensive or traceable database for reference, and the best opportunity to gather the data is to collect them in their published year. All the data analyzed in this study were originally collected in their published year by our supporters who worked in Employment Guidance Centre in universities. However, their purpose in collecting the data was mainly for employment guidance and not directly for this study. As a retrospective collection, the year of the data of each corporation was not continuous.

Furthermore, CASIC suspended open access to their comprehensive HR needs for years. On the contrary, CASC annually publishes their HR requirement online (http://www.spacetalent.com.cn/zhiweicx.aspx); thus, the data quality is more desirable. I have cited the address in the manuscript for reader’s reference (line 185). As mentioned above, however, there are only available data (in Chinese language) of the current year.

To Major Issue 3:

I regret the ambiguities. In the table, the words “Number of x-tier disciplines” means that the number of disciplines in the tier, not its code. I have improved the expression of 4.1 and hope it is now clear enough for readers. If an individual wishes to match the first-tier catalogue in Table 1 to “undergraduate major of general institution of higher learning”, they can click on the web of MOE (http://old.moe.gov.cn/publicfiles/business/htmlfiles/moe/s3882/201210/143152.html), as provided in the reference, and download the first appendix document online (however, it is in Chinese). In this document, one can find all of the first-tier disciplines (two-digital coded) of Table 1 in Chinese. I have translated and marked the first-tier discipline in the document that was added to the supplement file for your review. 

The catalogue of bachelor’s and postgraduate degrees was not completely identical in the MOE’s documents, and in the job posting the term of discipline used by the corporations and their subordinate units were irregular in many cases. It was difficult to classify all of the disciplines required by the corporations across different degrees, different units, and different years by simply adopting the MOE’s document. As a result, we developed the comprehensive three-tier coded classification system, based on the MOE’s documents (reference 32 and 33) with the assistance of an expert panel. During the panel’s work, the experts also reviewed the Chinese National Standard of Discipline Classification and Code (GB/T 13745-2009) for supplementation (https://dar.cwnu.edu.cn/__local/D/67/2D/71E79C7A1FDE42DF9B93E9303A7_7518C48A_611CB.pdf?e=.pdf). The new system was chiefly based on the first and second tiers’ structure in the document “undergraduate major of general institution of higher learning”. For the third-tier discipline, it was established principally based on “Check list of undergraduate specialties of general institution of higher learning” (the 2nd appendix of reference 33) and adjusted by combining and refining the national standard and the elite universities’ graduate and post graduate recruit catalogue. According to the coded classification system, we classified all of the raw data into appropriate coded titles for analysis.

Limited to the length of the manuscript, however, the comprehensive three-tier coded classification system cannot be demonstrated in its full form. I have added it to the letter as a supplemental file for review. However, the entire basic study was in a Chinese language environment, and the data, materials, and movement’s document are only in Chinese. To accurately translate these basic files to English may be too time-intensive; thus, I had hoped that the reader would be able see the general structure. However, I would be happy to translate any part of the file you wish. 

Furthermore, although there are very few differences between the newest version and the former version of discipline classification, as a retrospective study, we do not recommend the use of the newest version because the terminology in years of corporations’ required discipline are largely based on the older version. Further, their HR office may have latency.

To Major Issue 4:

As you have detailed, the assertion is somewhat subjective without direct public evidence, such as ratio of production, investment, or HR allocation. In the manuscript, the word “scientific research” refers to the science pure for truth. Certainly, the corporations participated in some innovative scientific research projects; however, the researches’ direction is mainly science for state power. Fully considering your valuable comment, I have adjusted the words in the text (lines 483-487). Moreover, to clarify the nature of the corporations regarding their core mission and activities, which could be seen as indirect evidence, I’d like to offer some explanation.

First, we might see the nature of the two corporations through their histories. China’s space/missile industry has been under development for more than sixty years, which can be segmented into three periods: the PLA unit period, the ministries of the PRC’s state department period, and the state-owned enterprise period. The industry was founded in February 1956; the fifth research institute of the ministry of national defense was established in Beijing. Since 1965, China established eight ministries of machine building (MMB) industry, in which MMB No. 7 was assigned for the space/missile industry. The MMBs were not merely some bureaucratic offices in central government but a huge system that included R&D, production, and training organizations across the country. When the PRC entered the era of reform and opening up, the MMB was reorganized into general industry corporations and later divided into CASC and CASIC. The evolution of the top organizations of the China’s space industry is shown in Appendix Figure 2. From their history, we can understand their mission and activities’ tradition is weapon systems engineering R&D and production for national defense and state power.

Second, we might understand their current principal engagement and mission from their official declaration. According to their website (in English), CASC is mainly engaged in the research, design, manufacture, test and launch of space products such as launch vehicles, satellites, manned spaceships, cargo spaceships, deep space explorers, and space stations, as well as strategic and tactical missile systems. CASIC takes “empowering the army with science and technology and serving the nation with aerospace technology” as its corporate mission and engages in a strategic industry related to national security. It has established an R&D and production system for air defense missile weapon system, aerodynamic missile weapon system, solid launch vehicle, and space technology products. 

Third, the two corporations were born from MMB: although they changed the name and reformed to fit an enterprise style to meet international convention, the core structure and main mission has not changed significantly. For example, CASC has 8 large R&D and production complexes (Appendix Table 2), 11 specialized companies, 13 listed companies and a number of directly affiliated units. The core units are the production complexes, each called an “Academy,” all of which have a long history and tradition that can be traced back to the MMB era. These academies have detailed industrial divisions for space and missile system R&D and production, which can be summarized based on open accessed information such as their websites. Academy No. 1, No. 5, and No. 8 are system integrators, while No. 4, No. 6, No. 7, and No. 9 are supporting units for subsystem or volume production (Appendix Fig. 3). No. 11 is a newly divided unit mainly for unmanned aerial vehicle. The series of their number is not continuous due to the splitting in 1993, and CASC did not renumber its Academies’ code.

By simply calculating three years’ (2019, 2017, and 2016) HR requirement of CASC, we could understand their (higher educated) manpower distribution (Appendix Fig. 4). Academy No. 1, No .5, and No. 8 account for 48.08% of manpower, while all the core units account for 85.90%. 

Generally speaking, to some extent, the corporations are space and weapon system engineering organizations by their nature. Although they have many scientific research activities, compared with their main mission, they are relatively insignificant. Due to the fact that their scientific activities are often publicly reported, some observational bias may be present in the outside world as there is a lack of transparency of many of the organizations’ principal missions.

To Minor Issue 1:

I quite agree with you on this point. Originally, I had detailed the history and structure of the sample corporations; however, considering the length of text, it had been simplified. I have therefore inserted relevant information into section 3 (lines 162-166). 

To Minor Issue 2:

Thank you for your correction; this has been amended.

To Minor Issue 3:

Thank you for your reminder. When comparing the mean of two independent samples, it is important for homogeneity of variance. Only the factor of Patents and Tradition’s equal variances are assumed; however, in the cases where equal variance was not assumed, t’-tests were used to adjust the results, which show that the significances have not changed. Considering the equivalence of t-test and ANOVA in the two independent samples’ test, we have adjusted the method to t-test in the manuscript to simplify the expression (lines 377-378, 448-450).

To Minor Issue 4:

Thank you for this suggestion. I have provided names to the four clusters (lines 546, 550, 556, and 562).

To Minor Issue 5:

To polish the language, the manuscript has been edited by an academic language editing company Editage (www.editage.cn) at each stage of submission. 

For Table 5, all the university’s names in the table are their official name. However, I have used the former official name for “Beijing University of Aeronautics and Astronautics” because its current official name “Beihang University” is harder to understand for English readers. It is possible that these names have not been translated perfectly; however, considering that the reader could search for them online in English, in my opinion, the official names should be kept.

To Minor Issue 6:

I have understood the request. I will consider the option, thank you.

We have put our best efforts into revising and improving the manuscript. These changes will not influence the content and framework of the paper. We have listed the changes within this document, and have marked them in red in the revised paper. We appreciate your considerate remarks, and hope that the corrections will meet with approval.

Once again, thank you very much for your comments and suggestions.

Sincerely yours,

Xiao-xiao Li

Department of Education and Research on National Defence, Fuzhou University

No. 2, Xueyuan Road, Minhou County, Fuzhou 350116, P. R. China

Tel: 86 15001338803

Email: shawnleemm@gmail.com; shawnlee@fzu.edu.cn

(Appendix in the file ‘Rebuttal_letter’ which we uploaded)

---

## [Editor Report · Decision Letter 1]

23 Nov 2020

Support potential of elite civil universities for China’s space industry: higher educational mobilization capacity for China’s space ambition

PONE-D-20-25196R1

Dear Dr. Li,

We’re pleased to inform you that your manuscript has been judged scientifically suitable for publication and will be formally accepted for publication once it meets all outstanding technical requirements.

Kind regards,

Bing Xue, Ph.D.

Academic Editor

PLOS ONE
---

## [Editor Report · Acceptance letter]

1 Dec 2020

PONE-D-20-25196R1 

Support potential of elite civil universities for China’s space industry: higher educational mobilization capacity for China’s space ambition 

Dear Dr. Li:

I'm pleased to inform you that your manuscript has been deemed suitable for publication in PLOS ONE. Congratulations! Your manuscript is now with our production department. 

Kind regards, 

on behalf of

Professor Bing Xue 

Academic Editor

PLOS ONE